# Phosphorylation, Mg-ADP, and Inhibitors Differentially Shape the Conformational Dynamics of the A-Loop of Aurora-A

**DOI:** 10.3390/biom11040567

**Published:** 2021-04-12

**Authors:** Zahra Musavizadeh, Alessandro Grottesi, Giulia Guarguaglini, Alessandro Paiardini

**Affiliations:** 1Department of Biochemical Sciences “A. Rossi Fanelli”, Sapienza University of Rome, 00185 Rome, Italy; 2Super Computing Applications and Innovation, CINECA, 00185 Rome, Italy; alegrot@gmail.com; 3Department of Biology and Biotechnology, Institute of Molecular Biology and Pathology, National Research Council, Sapienza University of Rome, 00185 Rome, Italy; giulia.guarguaglini@uniroma1.it

**Keywords:** Aurora-A, molecular dynamics simulation, activation loop, conformational dynamics

## Abstract

The conformational state of the activation loop (A-loop) is pivotal for the activity of most protein kinases. Hence, the characterization of the conformational dynamics of the A-loop is important to increase our understanding of the molecular processes related to diseases and to support the discovery of small molecule kinase inhibitors. Here, we carry out a combination of molecular dynamics (MD) and essential dynamics (ED) analyses to fully map the effects of phosphorylation, ADP, and conformation disrupting (CD) inhibitors (i.e., CD532 and MLN8054) on the dynamics of the A-loop of Aurora-A. MD revealed that the stability of the A-loop in an open conformation is enhanced by single phospho-Thr-288, while paradoxically, the presence of a second phosphorylation at Thr-287 decreases such stability and renders the A-loop more fluctuant in time and space. Moreover, we found that this post-translational modification has a significant effect on the direction of the A-loop motions. ED analysis suggests that the presence of the phosphate moiety induces the dynamics of Aurora-A to sample two distinct energy minima, instead of a single large minimum, as in unphosphorylated Aurora-A states. This observation indicates that the conformational distributions of Aurora-A with both single and double phospho-threonine modifications are remarkably different from the unphosphorylated state. In the closed states, binding of CD532 and MLN8054 inhibitors has the effect of increasing the distance of the N- and C-lobes of the kinase domain of Aurora-A, and the angle analysis between those two lobes during MD simulations showed that the N- and C-lobes are kept more open in presence of CD532, compared to MLN8054. As the A-loop is a common feature of Aurora protein kinases, our studies provide a general description of the conformational dynamics of this structure upon phosphorylation and different ligands binding.

## 1. Introduction

The Aurora protein kinases are attractive targets for the development of new therapeutics, for their involvement in processes associated with the progression of several cancers. Aurora kinases are classified into three sub-families consisting of Aurora-A, Aurora-B, and Aurora-C, with high homogeneity in mammalian cells [1]. Aurora-A regulated processes include centrosome maturation and separation, followed by assembly of a bipolar spindle, a trigger of mitotic entry, alignment of chromosomes in the metaphase, cytokinesis/abscission, and proteolytic degradation. Destruction of Aurora-A leads to cell cycle arrest, while overexpression has been found in many cancer cell lines [2,3].

The kinase domain of Aurora-A is composed of an N-terminal (N-lobe) and a C-terminal region (C-lobe), which are connected by a flexible hinge region (Figure 1). A deep cleft is present between these two lobes, in which a molecule of ATP or ADP is bound to one or two divalent Magnesium cations, which compensate for the strong negative charge of the ATP/ADP phosphates [4]. The N-terminal part of this domain consists of a five-stranded β-sheet and an α-helix, termed the “C-helix”. Aurora-A has also a short α-helix called the “B-helix”, which is just prior and perpendicular to the C-helix. The glycine-rich segment lies at a sharp turn that joins two antiparallel strands at the beginning of the β-sheet (β1-β2). The C-terminal lobe consists of seven α-helices and a four-stranded β-sheet and contains the catalytic aspartic acid of the HRD motif (sequence His-Arg-Asp at positions 254–256) and the mobile A-loop, whose position and conformation determine whether a kinase is active or inactive. The Aurora A-loop spans residues 274 to 299, beginning with a DFG (Asp-Phe-Gly) sequence motif that is highly conserved among kinases and ending with an APE (Ala-Pro-Glu) motif (sequence PPE in Aurora-A) [5] (Figure 1).

Aurora-A has conserved sequence and structural features that are organized into an “open” conformational arrangement, whenever the kinase is in an active state. In contrast, such features are displaced and unorganized in a “closed” state, when the kinase is inactive (Figure 2). The A-loop and the control of its conformation, with ligands and post-translational phosphorylation, play a key role in modulating the activity of Aurora-A. Active/open forms of Aurora-A are phosphorylated on Thr-287 and Thr-288 of the A-loop (Figure 1). The phosphorylation of Thr-288 is known to activate the kinase, but the function of Thr-287 phosphorylation remains unclear [6,7]. In its unphosphorylated form, Aurora-A has been captured in an inactive/closed conformation in which the C-helix and the A-loop are sometimes disordered and/or rearranged (Figure 2). Therefore, the fundamental differences between the active/inactive states of Aurora-A are essentially determined by inter-lobe motions, the position of the C-helix, which is twisted outward from the N-lobe in the closed state, and the A-loop that is displaced onto the C-terminal lobe, as shown in Figure 2.

In recent last years, several studies revealed that the fully active/open conformational state of Aurora-A is physically associated with the product of the *MYCN* oncogene, and thereby protects N-Myc from proteasomal degradation [8]. Indeed, the Aurora-A/N-Myc interaction is critical for stabilizing the latter, resulting in high levels of expression of the oncoprotein, which in turn drives aggressive forms of human tumors, including neuroblastoma [9]. In spite of its pivotal role in cancer occurrence and aggressiveness, N-Myc is not easily druggable due to the lack of clefts or site surfaces for binding of small molecules [10,11]. Hence, targeting N-Myc proteins has been challenging with the current approaches. The development of a new class of inhibitors has been recently identified that act by binding the ATP binding pocket of Aurora-A, to drive closed/inactive conformational changes that prevent Aurora-A/N-Myc interaction, inducing the degradation of the oncoprotein [12]. These conformation-disrupting (CD) inhibitors, i.e., CD532 (PDB ID: 4J8M) [13] and MLN8054 (PDB ID: 2WTV) [14], drastically alter the conformation of the A-loop, G-loop, and C-helix of Aurora-A in a tightly closed state, compared to either the open state of Aurora-A in the apo- form (empty binding pocket) or complexed with the conventional non-CD inhibitors [15]. Here, we set out to investigate in detail the main conformational changes of Aurora-A in different states, by using molecular dynamics (MD) and essential dynamics (ED) simulations of the kinase in the presence of different ligands, including MLN8054 and CD532 in the closed state, and ADP, along with different phosphorylation states of Thr-287 and Thr-288, in an attempt to explain how the dynamical behavior of the kinase changes in the different states and conformations.

## 2. Materials and Methods

### 2.1. MD Simulations

The starting structures of Aurora-A used for MD and ED simulations were taken from the Protein Data Bank (PDB) (www.rcsb.org/pdb, accessed on 8 April 2021) [16]. In particular, the following structures were selected: (1) Apo-Aurora-A (PDB ID: 4J8N) [13]; (2) Aurora-A in complex with ADP in the presence of two Magnesium ions, without phosphorylated threonine residues (PDB ID: 1MQ4) [17]; Aurora-A in complex with ADP, in the presence of two Magnesium ions, with (3) one (Thr-288) or (4) two phosphorylated threonine residues (287–288) (PDB ID: 1OL7) [18]; (5) Aurora-A in presence of CD532 (PDB ID: 4J8M) [13]; (6) Aurora-A in presence of MLN8054 (PDB ID: 2WTV) [14]. All the selected structures have no more than two residues missing continuously in any region of the kinase domain. In the case of PDB 2WTV, three mutations were performed (i.e., Arg215Leu, Glu217Thr, and Lys220Arg) to obtain the wild-type kinase protein. The missing residues and mutations were modeled using the Pymod3 software for the selected X-ray structures [19]. MD simulations were performed with GROMACS 2018.1 [20] (www.gromacs.org, accessed on 21 March 2018) with a modified version of the CHARMM-36 force field [21] for the protein, including parameters for the phosphorylated tyrosines. The ligand parameters were taken from the CHARMM General Forcefield [22]. The structures were centered in a cubic box with a minimum distance of 0.7 nm between the solute and the box boundary. The box was then filled with TIP3P water molecules [23]. The ionic strength was adjusted to make sure all simulations were electrically neutral, parameters for counterions were taken from CHARMM-36 force field. Energy minimization was executed by the steepest descent method and the conjugate gradient method for the subsequent 50,000 steps. Nonbonded forces were modeled using the particle-mesh Ewald (PME) method with a cutoff distance of 12 Å, used for all simulations. Equilibration of starting structures was performed by a procedure where position restraints were applied in a stepwise manner to protein and ligands in the complexes with a force constant of 1000, 500, and 250 kJ/mol/nm. The initial velocities were taken randomly from a Maxwellian distribution at 300 K. The temperature was held constant (300) K [24]. Long-range electrostatic interactions were calculated using the PME summation methods [25]. Lennard–Jones interactions were calculated using a cutoff of 12 Å. The pair lists were updated every 1 step. The LINCS algorithm [26] was used to constrain bond lengths. The time step was 2.0 fs, and coordinates were saved every 1.0 ps.

### 2.2. Principal Component Analysis

To study the collective motions of the kinase, an MD simulation at 300 K was performed for 100 ns. From the equilibrated portion of the trajectory (beyond 10 ns), totaling 4000 frames for each system, the covariance matrix of the positional fluctuations of the Cα carbon atoms was built up and diagonalized. The procedure yielded new axes (eigenvectors), representing the directions of the concerted motions. The corresponding eigenvalues gave the mean-square positional fluctuation for each direction [27]. We projected the trajectory to the largest five eigenvectors to identify detailed structural information. To calculate the angle between two lobes, 14 residues in each lobe have been considered as vectors, then the angle between these two vectors has been calculated using the dot product method along the simulation time. The principal components analyses have been performed using the *covar* routine available in the Gromacs suite and the projections onto the eigenvectors have been calculated using the *anaeig* routine, still available in the Gromacs suite (www.gromacs.org, accessed on 21 March 2018).

### 2.3. Cluster Analysis

Cluster analysis was performed on the equilibrated portion of all trajectories (beyond 10 ns) to find the best representative of the A-Loop of the trajectories sampled by Aurora-A. The analysis was performed according to the algorithm described in Daura et al. [28]. Briefly, the algorithm counts the number of neighbors using a cut-off of 0.1 nm, then takes the structure with the largest number of neighbors with all its neighbors as a cluster and eliminates it from the pool of clusters. The process is then repeated for the remaining structures in the pool. For all systems, a total of 4500 frames were analyzed and clustered. For each system, the first three representative clusters were used for the analysis of the A-Loop conformations.

### 2.4. Inter-Lobe Angle Analysis

The N- and C- inter-lobes angle was measured by calculating the dot product between the normal of planes that best fitted the N- and C-lobes. Basically, a plan spanning the residues in the N- and C-lobe have been defined by fitting the C-alpha carbon atoms of residues encompassing the two lobes. Then, the norm of each plane has been calculated and the dot product of the two norms defined the inter-lobe plane. Only the C-alpha carbon of the N- and C-lobes was used for fitting. We have used the *gangle* routine in Gromacs (www.gromacs.org, accessed on 21 March 2018).

## 3. Results

A total of six simulations, 100 ns each, of Aurora-A in different conditions were run. Two structures were in the closed/inactive state, in the presence of inhibitors CD532 and MLN8054. Four structures were in the open/active state: Aurora-A without ligands (apo Aurora-A), Aurora-A in complex ADP/Mg ions, Aurora-A in complex with ADP/Mg and with two phosphothreonines (Thr-287, Thr-288; 2TPO) and Aurora-A in complex ADP/Mg with a single phosphothreonine (Thr-288; TPO). To explore the dynamic stability of the complexes during the simulation, the root mean square deviation (RMSD) of protein backbone atoms and potential energy (Appendix A) of the system were calculated. To this end, the RMSD was calculated and plotted in Figure 3. As the figure shows, the RMSD of all six systems in the 100 ns range provide evidence that all the six simulated systems have reached a local potential energy minimum and a state of equilibrium was achieved for all the molecules after 10 ns, which shows the patterns of convergence of the RMSD trajectory. The active states (Figure 3a) showed the least RMSD value converging between 0.2 and 0.3 nm. In comparison with the active states of the kinase, the two inactive states showed lower deviating patterns.

The RMSD plot of the activation loop suggests that the A-loop is stabilized when Aurora is complexed with ADP/Mg and in presence of a single TPO, in comparison to all the active states. It is worth noting that the conformational stability of the A-loop is enhanced by phosphorylation at position 288 alone, while it is decreased when both positions 287 and 288 are phosphorylated (2TPO, Figure 3b). The inactive proteins in complex with CD532 and MLN8054 showed convergence, with an RMSD value around 0.2 nm at 100 ns, with similar stability during the simulation (Figure 3c). To analyze the extent of the A-loop stability over the simulations, we have plotted the RMSD of the A-loop of Aurora-A for closed systems, and the results indicated high stability of the A-loop in presence of MLN8054 (Figure 3d). This aspect has been also analyzed by carrying out a cluster analysis of the A-loop conformations sampled throughout the MD simulations (see Methods for details). The statistics of the cluster analysis have been summarized in the Appendix A. The three most representative conformations of this clusterization (see Appendix A) provide further evidence that the A-loop in the inactive states (in the presence of CD532 and MLN8054) is stabilized in a closed conformation throughout the MD simulation, while the A-loop in the active states sample different stable conformations. Appendix A reports the total number of cluster members for each cluster and the number of cluster members transitions as well. As the table shows, typically, the total number of clusters detected was in the range of 20–55 in total, with the only exception of Aurora-A state in the presence of CD532, where a total of 248 clusters were detected. Despite this large variability, the superimposition of the centroids of the first three clusters of Aurora-A in the active state (Appendix A) provide evidence that the sampled A-loop conformations were comparable. For Aurora-A in the inactive state (Appendix A), the superimposition of the first three centroids shows that A-loop displays a larger variability of the conformations, all compatible with an Aurora-A state in the closed conformation. In addition, the number of cluster transitions of the first three clusters (Appendix A) suggests that the conformations sampled by the members of the first three clusters are very similar (particularly in the active state).

### 3.1. Collective Motions

Residue-based root means square fluctuation (RMSF) values of the systems, starting at 10 ns, were calculated to compare the flexibility of each amino acid residue of the complex. Furthermore, this RMSF plot showed the flexible parts of the protein (Figure 4). Higher RMSF values indicate greater flexibility during the MD simulation. The RMSF plot indicated that the A-loop is the most flexible region in all the simulations, as expected, except for terminal parts of the protein (Figure 4a). In particular, using the A-loop region to fit the MD trajectories prior to the calculation, the RMSF plot of the A-loop displayed that the apo- system was highly fluctuating in the A-loop, and precisely at two Arginine residues (285–286), with an RMSF value of 0.4 nm. The Aurora-A structure in complex with ADP-Mg and in presence of a single TPO showed slight fluctuations in the A-loop, with the RMSF value less than 0.2 nm (Figure 4b). The fluctuation of the A-loop was enhanced by the phosphorylation at positions 287 and 288 (2TPO), while the A-loop is stabilized by the single phosphorylation at position 288, and resulted in a more active/open kinase, compared to 2TPO. The A-loop in the closed states showed various fluctuating patterns throughout the simulation, in presence of CD532 and MLN8054. The RMSF plot in both closed states indicated high fluctuations in the αEF/αF loop (residues 302–306), with values reaching up to 0.3 nm, in comparison with all the open structures (Figure 4a). Besides, there were higher peaks of fluctuations observed in the C-helix in the case of MLN8054, and the A-loop in the case of CD532, reaching up to 0.35 nm (Figure 4c).

### 3.2. Characterizations of Global and Local Concerted Motions in Aurora-A

One of the purposes of this study is to characterize global vs. local dynamics in Aurora-A, to detect conformational changes upon ligand binding, and correlate global dynamics with local ligand binding effects. To this end, we used a statistical approach to study the role of the positional fluctuations in Aurora-A, in the presence and without inhibitors. Essential dynamics (ED) of MD trajectories is a method based on principal component analysis (PCA), for analyzing the conformational space sampled during the simulations, which can provide insights into partially populated conformational states relevant to catalysis. We, therefore, carried out PCA on Aurora-A open states, both unphosphorylated and phosphorylated, in addition to the closed states of Aurora-A in presence of CD532 and MLN8054, to highlight the differences of Aurora-A collective motions in these different systems. As illustrated in Figure 5, the simulated conformations in each system were dynamic and fluctuant during 100 ns MD simulations, and eventually stabilized into a dominant state. As shown in Figure 5a, the conformational changes of Aurora-A with both TPO and 2TPO were greater than those of the apo form and Aurora-A with the unphosphorylated A-loop. Indeed, the presence of the phosphate on the threonine residues of the A-loop induces the dynamics of Aurora-A to sample two distinct minima, instead of one single large minimum, as in the other Aurora-A active states. This suggests that the conformational distributions of Aurora-A with both single and double phospho-threonines were remarkably different from the unphosphorylated states (apo-Aurora-A and Aurora-A/ADP). The inactive systems, including Aurora-A/CD532 and Aurora-A/MLN8054, showed a different protein conformational state, suggesting that the inhibitors could undergo dramatic conformational fluctuations of Aurora-A in comparison to the active states (Figure 5b).

### 3.3. Ligand Interactions and Inter-Lobe Motions

In the analysis of protein dynamics, an important goal is the description of slow large-amplitude motions. Only global collective motions can significantly change the exposed surface of the protein and hence influence interactions with its environment. Essential modes of domain motions in Aurora-A were obtained from principal component analysis (PCA). The principal component analysis of our current work identifies the first components that represent the dominant motions of Aurora-A protein and the latter were plotted and compared for all the Aurora-A kinases. The results are shown in Figure 6. The first principal component characterized the A-loop motion, and the hinge bending motion between the N- and C-lobes.

In detail, apo-Aurora-A, and Aurora-A in presence of ADP/Mg with the unphosphorylated A-loop, showed an increase in the amplitude of the motion with the two lobes being closer to each other (Figure 6a). In other words, the distance of N- and C- lobes of Apo form decreases, and the A-loop shifts toward the N-lobe. The TPO and 2TPO Aurora-A revealed a different mode of action, which causes the A-loop to go in the opposite direction in order to preserve the open form of A-loop, in comparison to the unphosphorylated states (Figure 6a). At the global kinase domain level, the conformational dynamics of the Aurora-A depend greatly on the conformation of the A-loop, which is closed when CD-inhibitors such as CD532 and MLN8054 are bound.

In all the open states, the observed outgoing movement of the glycine-rich loop towards the outer surface during MD, together with the extended conformation of the A-loop, is contributing to the open conformation of Aurora-A that is required for substrate accessibility. On the contrary, in the closed states, binding of the CD532 and MLN8054 ligands has the effect of increasing the distance of the N- and C-lobes, although at different amplitudes (Figure 6b).

In the presence of MLN8054, all the residues that correspond to the αC helix region move away from the A-loop with higher amplitude and favor an outward movement, while in the presence of CD532, higher amplitudes during the MD were seen for the A-loop (Figure 6b). Taken together, the major structural transitions of the αC helix and the A-loop, and the movement of the αC helix outwards the activation segment, ultimately led to the sufficient space available to accommodate the inhibitors CD532 and MLN8054 between the N- and C-lobs in the active site, as seen in the closed state of Aurora-A.

### 3.4. Angle Analysis

To assess whether the observed opening of the kinase lobes [15] by CD532 and MLN8054, compared to the apo-kinase, is maintained during the MD simulations, an angle analysis between the N- and C-lobes of the kinase was carried out. The effect of increasing the distance of the N- and C-lobes appears evident also analyzing the angle between the two lobes in presence of CD532 and MLN8054 ligands, as compared to the apo-Aurora-A state. The results show that the N- and C-lobes of Aurora-A in presence of CD532 and MLN8054 are widely more open across the whole MD simulation than in the apo form (Figure 7a). The mean angle value between the lobes in presence of CD532 was ~80.9°, while the same value for MLN8054 was about ~79.1° (Figure 7b).

Moreover, angle analysis for all open states was carried out between N- and C-lobes in different conditions, the mean angle value between the two lobes in presence of ADP/MG, ADP/MG with 2TPO and single TPO was ~77°, while the same value for Apo was about ~74° (Appendix A).

### 3.5. Protein-Ligand Contacts Analysis

Finally, we have measured the number of contacts between the Aurora-A A-loop and the inhibitors CD532 and MLN8054 as a function of simulation time and plotted the results in Figure 8. It is evident that the interaction between Aurora-A and MLN8054 results in a larger number of protein-ligand contacts (28 ± 4) as compared to the corresponding value for CD532 (8 ± 2).

## 4. Discussion

In this study, we reported the dynamical characterization of the A-loop of Aurora-A in different conditions, covering a wide range of states of the A-loop, from fully open, to fully closed (Figure 9).

Four active structures of Aurora-A in different states, addressing the effect of site-specific phosphorylation (Thr-287 and Thr-288) on the conformation of the A-loop, were initially investigated [29]. The A-loop is mostly stabilized by the single phosphorylation at position Thr-288 and resulted in wider access to the active site and presumably a more active kinase, compared to the dual phosphorylation Thr-287 and Thr-288. This is rationalized by competition between phosphorylated Thr-287 and Thr-288 for a binding cleft composed mainly of arginine residues (Arg-180, Arg-255, and Arg-286) [30]. Our analysis is in agreement with experimental data, which suggests that the single phosphorylation at Thr-288 is the most favorable, resulting in the highest catalytic activity [31]. An inhibitory role for phospho-Thr-287, when present along-side phospho-Thr-288, is therefore confirmed. Based on previously solved crystal structures, it was not clear how Thr-287 phosphorylation might influence the activity of Aurora-A. Our analysis on the dynamics of the A-loop suggests that Thr287 phosphorylation acts like a compensatory regulatory mechanism to Thr288 phosphorylation, in order to destabilize the fully open state during Aurora-A activation [32].

Residue-based RMSF analysis showed a high fluctuation in the αEF/αF loop in closed structures in comparison with open structures, and in support of this, the αEF/αF loop conformation is often coupled to changes in the A-loop from an active to inactive state. The critical location of the αEF/αF loop and extensive interactions of this loop with the A-loop indicate that it plays a key role in the A-loop function. In kinase proteins that require phosphorylation, the αEF/αF loop could stabilize the active conformation of the phosphorylated A-loop and destabilize the unphosphorylated A-loop, preventing it from folding into an active conformation [33].

Moreover, we monitored the significant structural changes and the direction of motions in the inactive Aurora-A, in the presence of the CD-inhibitors CD532 and MLN8054. The results of the angle analysis confirmed that both compounds are able to affect the conformational dynamics of Aurora-A by increasing the distance between N- and C-lobes, but while CD532 keeps the two lobes widely open during the MD simulation due to a bulkier effect and deeply entering the binding site cleft, in contrast MLN8054 obtains the same effect via tight hydrophobic interactions with the A-loop over time, compared to CD532. [12] Therefore, taken together, these observations suggest that the CD inhibitors of Aurora-A can act by adopting two complementary, but distinct strategies: stabilizing the A-loop in a closed conformation via the formation of direct contacts, and/or wide-opening the N-terminal lobe of the kinase to create an ideal cleft for A-loop closure. Due to the importance of CD inhibitors of Aurora-A in preventing the interaction with N-Myc, and the resulting potential applications in treating neuroblastoma [34], such different strategies could be exploited to design new CD inhibitors for therapeutic purposes.

In summary, this analysis provides new dynamical insights into the distinctive conformational changes associated with the different ligands in the active and inactive forms of Aurora-A. Moreover, since the described global motions have been reported to be general Aurora protein kinases features, this approach could be extended to provide useful information for the discovery of inhibitors with increased affinity and specificity for a specified conformational state, establishing also a new way to utilize MD simulations in drug discovery and design, which may be transferred to other potential drug targets in future.

## Figures and Tables

**Figure 1 biomolecules-11-00567-f001:**
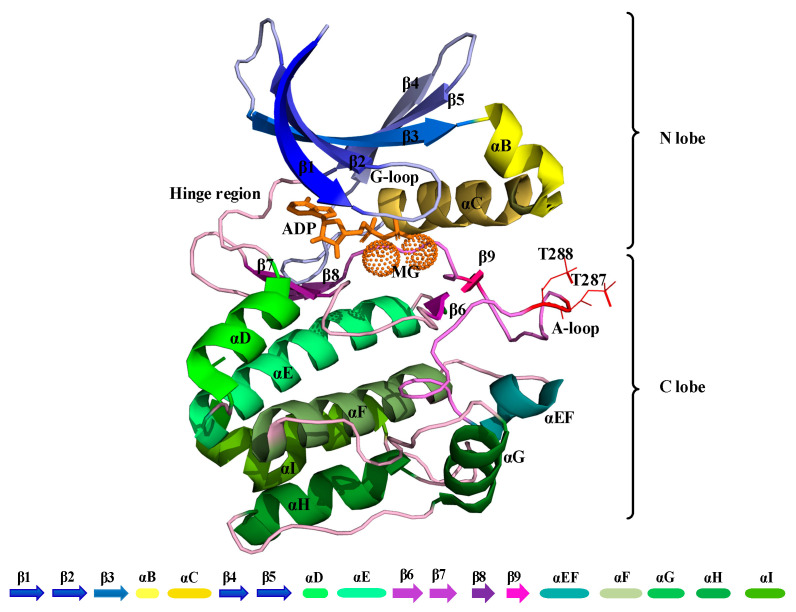
Cartoon representation of the kinase domain of Aurora-A (PDB 1OL7). The A-loop is colored light purple (residues 274–299) with the phosphorylated threonine residues at positions 287 and 288 shown as a line in red color. ADP is colored orange with coordinated Magnesium ions shown as light orange pointed spheres.

**Figure 2 biomolecules-11-00567-f002:**
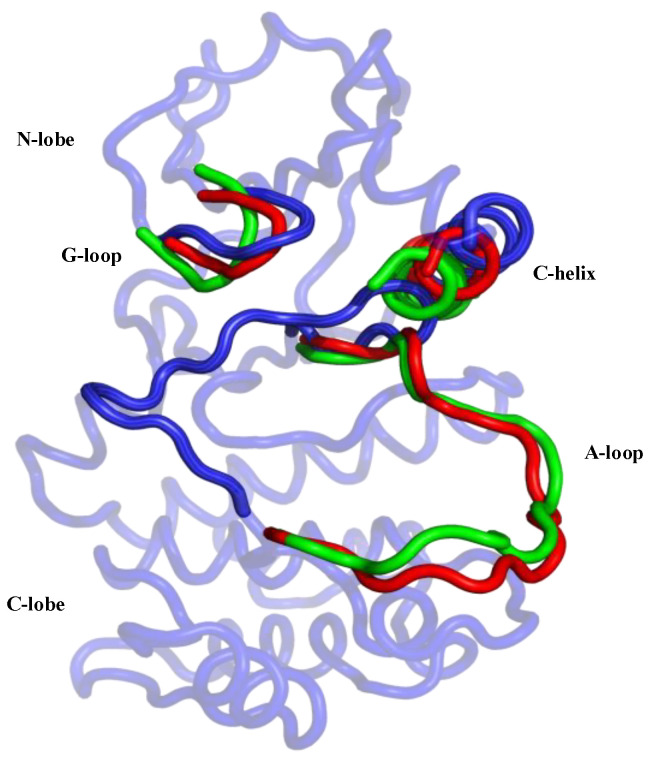
Graphical representation of the main differences between the two active/open states (phosphorylated/unphosphorylated) and the inactive/closed state of the Aurora-A protein. The closed conformation is represented in blue (PDB 4J8M) and the protein regions of the open forms in the phosphorylated and unphosphorylated states are colored in the green (PDB 1OL7) and red (PDB 1MQ4), respectively.

**Figure 3 biomolecules-11-00567-f003:**
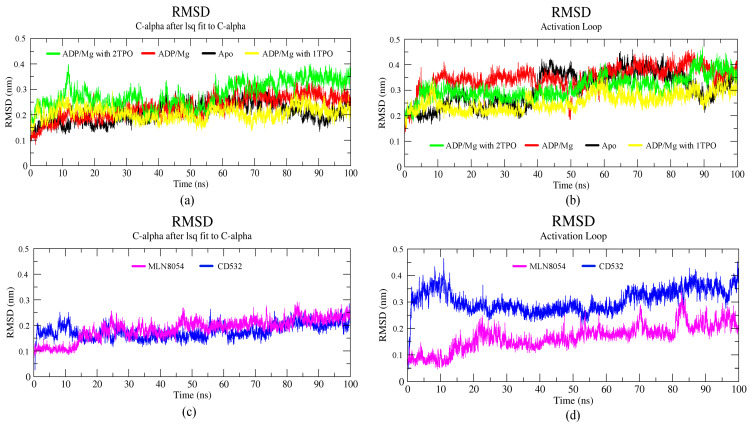
Backbone root mean square deviation (RMSD) of the Aurora-A protein for 100 ns MD. (**a**) RMSD plot of open/active states. (**b**) RMSD value of the A-loop in the active states: apo (4J8N, black), ADP/Mg (1MQ4, red), ADP/Mg with 2TPO (1OL7, green), ADP/Mg with single TPO (1OL7, yellow). (**c**) RMSD plot of inactive states. (**d**) RMSD value of A-loop in the inactive states: CD532 (4J8M, blue), MLN8054 (2WTV, pink).

**Figure 4 biomolecules-11-00567-f004:**
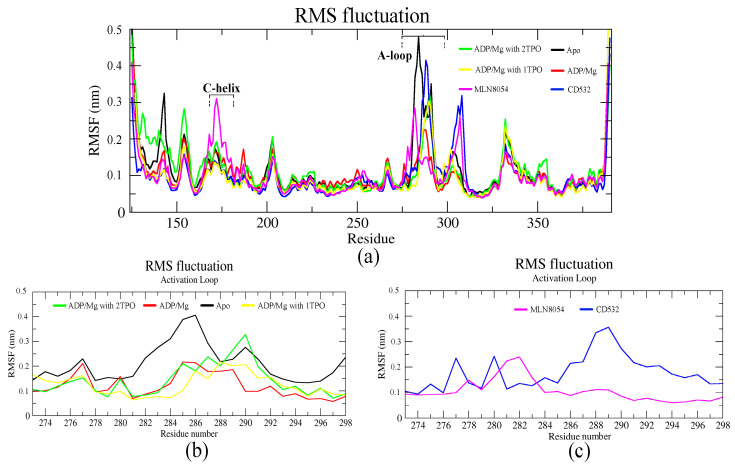
The RMSF plots of the Aurora-A protein for 100 ns MD. (**a**) RMSF plot of all systems including active and inactive states. (**b**) RMSF value of the A-loop in the active states: apo (4J8N, black), ADP/Mg (1MQ4, red), ADP/Mg with 2TPO (1OL7, green), ADP/Mg TPO (1OL7, yellow). (**c**) RMSF value of A-loop in the inactive states: CD532 (4J8M, blue), MLN8054 (2WTV, pink).

**Figure 5 biomolecules-11-00567-f005:**
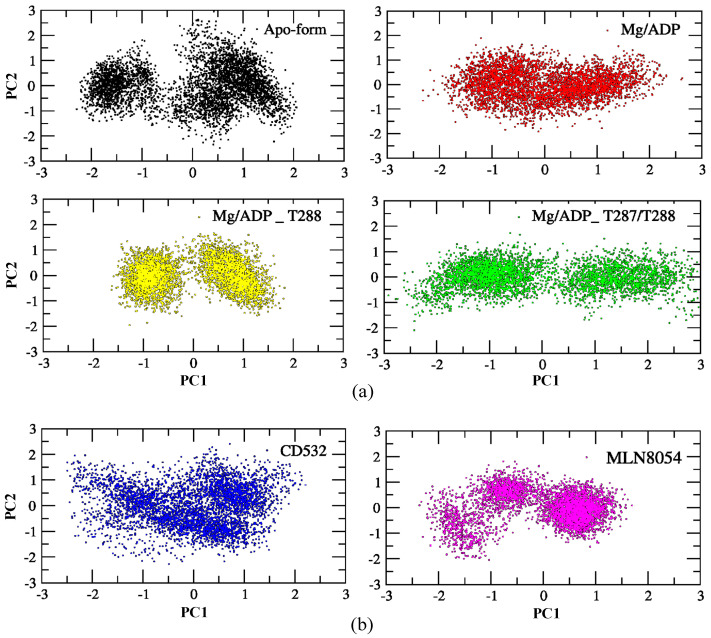
PCA scatter plot along with the first two principal components of the Aurora-A protein for 100 ns MD simulation. (**a**) the active structures: apo (4J8N, black), ADP/Mg (1MQ4, red), ADP/Mg with 2TPO (1OL7, green), ADP/Mg with TPO (1OL7, yellow). (**b**) the inactive states: CD532 (4J8M, blue), MLN8054 (2WTV, pink). The corresponding main representative structures of the domain areas of the PCA are shown in Appendix A.

**Figure 6 biomolecules-11-00567-f006:**
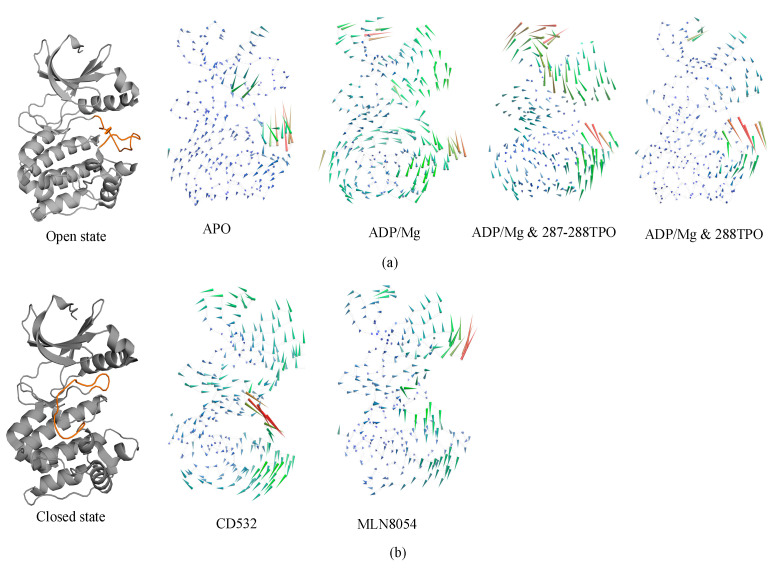
Porcupine plot showing the significant motions across the first PC. (**a**) the open/active states: apo (4J8N), ADP/Mg (1MQ4), ADP/Mg with two TPO (1OL7), ADP/Mg with single TPO (1OL7). (**b**) the inactive states: CD532 (4J8M) and MLN8054 (2WTV). The color of the protein strands is related to the extent of motion, and arrows show the direction of the correlated motion. Red reflects the highest movement followed by green, whereas blue depicts the least moving parts of the protein.

**Figure 7 biomolecules-11-00567-f007:**
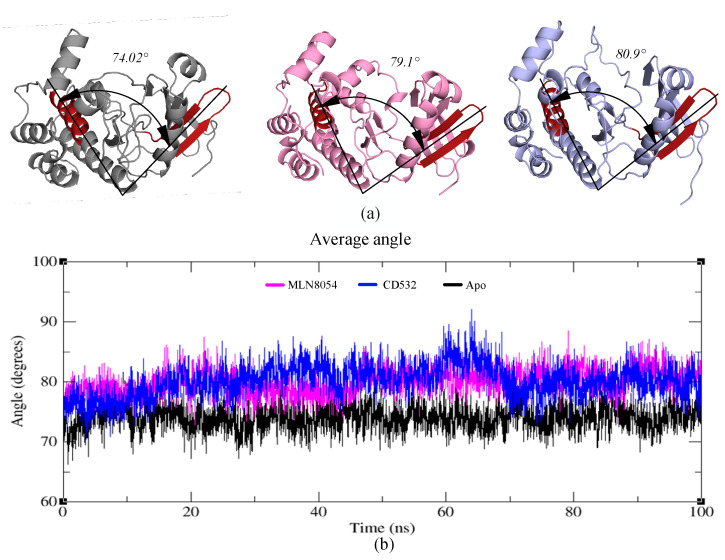
(**a**) Representation of the crystal structures of apo-Aurora-A (4J8N, black), Aurora-A with MLN8054 (2WTV, pink), and Aurora-A with CD532 (4J8M, blue). The angle between two vectors joining the α-carbons Leu196 to Leu210 (in the red hairpin), and Val310 to Val324 (red α-helix), is shown and labeled. (**b**) the plot represents the angle value during the trajectories of Aurora-A for 100 ns MD simulations.

**Figure 8 biomolecules-11-00567-f008:**
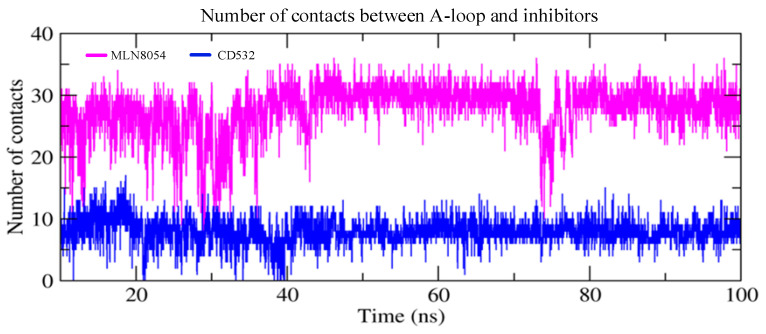
The number of contacts between the A-loop and the inhibitors as a function of simulation time in two closed/inactive structures including Aurora-A/MLN8054 (2WTV, pink), and Aurora-A/CD532 (4J8M, blue).

**Figure 9 biomolecules-11-00567-f009:**
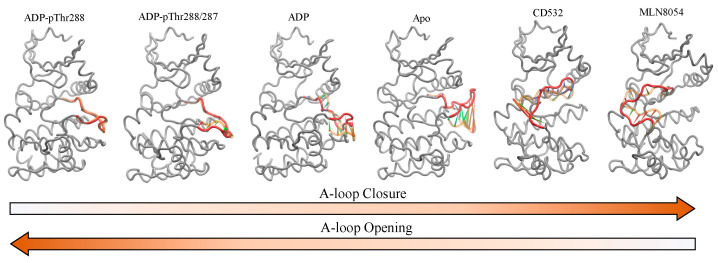
Porcupine plot showing the significant motions of the A-loop of Aurora-A, including the open/active states: ADP/Mg with single TPO (1OL7), ADP/Mg with two TPO (1OL7), ADP/Mg (1MQ4), apo (4J8N), and the closed/inactive states: CD532 (4J8M) and MLN8054 (2WTV). The A-loop is colored red (residues 274–299), and the first position of the A-loop is indicated in light orange. Arrows show the direction of the correlated motion. Red reflects the highest movement followed by green, whereas blue depicts the least moving parts of the protein.

## Data Availability

The authors confirm that the data supporting the findings of this study are available within the article and Appendix A.

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
