# Peer review of "Phosphorylation, Mg-ADP, and Inhibitors Differentially Shape the Conformational Dynamics of the A-Loop of Aurora-A"

_biomolecules, 2021, doi:10.3390/biom11040567_

Round 1

Reviewer 1 Report

The manuscript investigated the conformational dynamics of the A-loop in a kinase protein Aurora-A using Molecular Dynamics simulations. Specifically, authors analyzed the effects of phosphorylation, ADP, and conformation disrupting inhibitors on the dynamics of the A-loop in Aurora-A. The work is of potential interest to the readership of Biomolecules.  My major concerns are as following.

1) The A-loop structure in the complex of Aurora-A and MLN8084 (PDB 2x81) is completely missing. The modeling details of the A-loop in this structure should be provided. Because the A-loop is a long loop containing over 20 residues, it is challenging to model such long flexible loop. Authors should analyze the reliability of the modeled structure. The conclusions based on this modeled structure may need to be re-investigated.

2) 2.1 MD simulations. Parameters (such as force field) for ligands, phosphorylated residues, and ions need to be described in detail.

3) 2.2 Principal component analysis. In this study, the whole length of MD trajectories (100 ns) were used for PCA calculation. However, the RMSD results show that systems reached equilibrium after 10 ns (page #5, lines 163-164). Authors need to exclude snapshots from the first 10 ns for PCA calculations. In addition, more details of the calculation moethod are expected. For example, which programs did authors use for PCA calculation? How many snapshots were used for calculation?

4) page #5 lines 168-177, RMSD values here cannot reflect the stability of the structures. They are the values of deviation of snapshots from their initial structure. These RMSD values may tell the system reached equilibrium in MD simulation. Authors may also analyze the energy of the system during the MD simulation to see if they reached equilibrium states. In Figure 3b and d, significant changes of RMSD values were observed during the MD simulation. It would be interesting to cluster conformations of A-loop from the MD trajectory and show representative conformations.

5) Figure 4, RMSF values of residues in A-loop shown in Fig. 4(a) are significantly larger than those shown in zoom in figures 4 (b) and (c). Using apo structure as an example, the peak value is about 0.47 nm in Figure 4(a). However, the peak value in figure 4(b) is only 0.4 nm. Any explanation? In addition,  figures would be more readable if a label is added for each line in the figure. In addition to the A-loop, another interesting peak was observed for the region near residue 305. RMSF values of this region in two inactive states are significantly higher than those in other four active states. Any comments/discussions?

6) Figure 5, two distinct domains were observed for several systems. Is it possible to show a representative structure for each domain?

7) 3.4 Angle Analysis. Authors presented data for three structures, two with CD inhibitors and one is apo structure. How about the other three structures?

8) I do not get the purpose of the protein-ligand contacts analysis (page #9, lines 290-295). In addition, the data for structures containing the ligand ADP is missing.

9) Proposed model of the A-loop dynamics. Authors need to explain in detail why the proposed model is reasonable based on the results presented in the manuscript. Unfortunately, I did not see any explanation.  

Some minor issues:

1) Figure 3 and 4, values at y-axes, “,” should be “.”

2) Figure 3, 4, 7, 8, adding a label for each line will make the figures much more readable.

3) page #6, line 204, “TPO2” should be “2TPO”, check other typos in the manuscript.

4) The title in Figure 8, “Number of contacts < 0.4 nm” is incorrect.

Author Response

Dear Editor,

We wish to thank both referees for their constructive criticisms. We have revised our manuscript to accommodate as much as possible of their suggestions. We feel that, thanks to the referees’ suggestions, the manuscript is now highly improved and ready for the audience of Biomolecules. Our responses are as follows:

  • Referee #1

The manuscript investigated the conformational dynamics of the A-loop in a kinase protein Aurora-A using Molecular Dynamics simulations. Specifically, authors analyzed the effects of phosphorylation, ADP, and conformation disrupting inhibitors on the dynamics of the A-loop in Aurora-A. The work is of potential interest to the readership of Biomolecules.  My major concerns are as following.

1) The A-loop structure in the complex of Aurora-A and MLN8084 (PDB 2x81) is completely missing. The modeling details of the A-loop in this structure should be provided. Because the A-loop is a long loop containing over 20 residues, it is challenging to model such long flexible loop. Authors should analyze the reliability of the modeled structure. The conclusions based on this modeled structure may need to be re-investigated.

Actually, from the beginning of this analysis, we made use of PDB 2WTV. We are deeply sorry, but due to a typo on earlier versions of the methods section, we wrongly indicated PDB 2X81 in place of the right one, i.e. PDB 2WTV, as the structure of Aurora-A in presence MLN8054. In the case of PDB 2WTV, where the structure of the activation loop is clearly visible, three residues were mutated compared to the WT protein kinase domain (i.e., Leu215Arg, Thr217Glu and Arg220Lys). Such residues were therefore retro-mutated to obtain the wild-type kinase protein. We detailed this part in the methods section and modified the manuscript accordingly.

2) 2.1 MD simulations. Parameters (such as force field) for ligands, phosphorylated residues, and ions need to be described in detail.

We have modified the Methods section so as to include specific information for phosphorylated residues, ions and ligand.

3) 2.2 Principal component analysis. In this study, the whole length of MD trajectories (100 ns) were used for PCA calculation. However, the RMSD results show that systems reached equilibrium after 10 ns (page #5, lines 163-164). Authors need to exclude snapshots from the first 10 ns for PCA calculations. In addition, more details of the calculation method are expected. For example, which programs did authors use for PCA calculation? How many snapshots were used for calculation?

The principal components analysis, and other analyses as well, have been performed using only the equilibrated part of all simulations, that is, after 10 ns of equilibration. There is a specific sentence in the Methods section to highlight this aspect. We have also included information about the used software and frame statistics in the Methods.

4) page #5 lines 168-177, RMSD values here cannot reflect the stability of the structures. They are the values of deviation of snapshots from their initial structure. These RMSD values may tell the system reached equilibrium in MD simulation. Authors may also analyze the energy of the system during the MD simulation to see if they reached equilibrium states. In Figure 3b and d, significant changes of RMSD values were observed during the MD simulation. It would be interesting to cluster conformations of A-loop from the MD trajectory and show representative conformations.

We thank the referee for these valuable suggestions and, accordingly, we have calculated the potential energy of all systems as a function of time (Supplementary Figure S1) and performed a cluster analysis as well of the A-loop conformations sampled by all systems in the equilibrated part of all trajectories (10-100 ns). We have inserted a paragraph in the results section and added a new supplementary figure (Fig. S2)

5) Figure 4, RMSF values of residues in A-loop shown in Fig. 4(a) are significantly larger than those shown in zoom in figures 4 (b) and (c). Using apo structure as an example, the peak value is about 0.47 nm in Figure 4(a). However, the peak value in figure 4(b) is only 0.4 nm. Any explanation? In addition, figures would be more readable if a label is added for each line in the figure. In addition to the A-loop, another interesting peak was observed for the region near residue 305. RMSF values of this region in two inactive states are significantly higher than those in other four active states. Any comments/discussions?

We have checked this aspect and highlighted the differences of the fluctuations in the different systems in the A-loop region and in the 302-306 region as well. About the scale along the y-axis, they are actually slightly different because we have fitted the trajectories onto the subset of alpha carbon atoms of the Loop-A residues only (namely, 274-299). The corresponding trajectory is then more representative of the actual internal fluctuations of the Loop-A region. That indeed introduces a slight change in the RMSF values as compared to the corresponding one calculated using the normal trajectory fitted onto the whole Aurora-A sequence. We have inserted a sentence in the corresponding section of the RMSF analysis to highlight this.

6) Figure 5, two distinct domains were observed for several systems. Is it possible to show a representative structure for each domain?

An additional supplementary Figure S3 has been included to show the main representative structures of the domain areas of the PCA (when detected). Namely, we have reported the representative for ADP/Mg with 2TPO, ADP/Mg with TPO, Apo and the inactive state MLN8054.

7) 3.4 Angle Analysis. Authors presented data for three structures, two with CD inhibitors and one is apo structure. How about the other three structures?

The angle analysis was actually carried out only on the two Conformation Disrupting (CD) inhibitors CD532 and MLN8054, to assess whether the observed opening of the kinase lobes [12], compared to the apo-kinase, was maintained during the MD simulations. Our results indeed showed that the N- and C-lobes of Aurora-A in presence of CD532 and MLN8054 are widely more open across the whole MD simulation than in the apo form (Figure 7a). The purpose of this analysis is now more clearly stated in the revised manuscript.

8) I do not get the purpose of the protein-ligand contacts analysis (page #9, lines 290-295). In addition, the data for structures containing the ligand ADP is missing.

As discussed in the manuscript, according to the crystal structures CD532 keeps the two lobes widely open by deeply entering the binding site cleft, while in contrast MLN8054 obtains the same effect via tight hydrophobic interactions with the A-loop [12]. Therefore, we wanted to assess whether the CD inhibitors of Aurora-A can act by adopting two complementary, but distinct strategies: stabilizing the A-loop in a closed conformation via the formation of direct contacts, and/or wide-opening the N-terminal lobe of the kinase to create an ideal cleft for A-loop closure. Therefore, contact analysis suggested that MLN8054 keeps tight hydrophobic interactions with the A-loop over time, while CD532 keeps the A-loop in a closed conformation indirectly (only few tight contacts with the A-loop).

9) Proposed model of the A-loop dynamics. Authors need to explain in detail why the proposed model is reasonable based on the results presented in the manuscript. Unfortunately, I did not see any explanation.

We thank the referee for having highlighted this point and we changed the manuscript accordingly. Actually, it was not our intention to propose any new model on Aurora-A A-loop dynamics, but only to stress on the fact that we reported the dynamical characterization of the A-loop of Aurora-A covering a wide range of states of the A-loop, from fully open, to fully closed. We hope that this is more clearly stated in the revised manuscript.

Some minor issues:

1) Figure 3 and 4, values at y-axes, “,” should be “.”→Done. Please see Figure 3 and 4.

2) Figure 3, 4, 7, 8, adding a label for each line will make the figures much more readable. →Done. Please see Figure 3, 4, 7, 8.

3) page #6, line 204, “TPO2” should be “2TPO”, check other typos in the manuscript. →Done

4) The title in Figure 8, “Number of contacts < 0.4 nm” is incorrect. →Done

Reviewer 2 Report

In line 44 & 45 they state that ATP/ADP is bound to 1 or 2 Mg cations while in Fig 1 they show 2 Mg cations. Is there still uncertainty about ths? In line 100 they should add " in an attempt' betwenn Thr-288 and to. In line 93 they should change This to These. They should give a reference when they first mention MLN 8084 & CD 532.

Author Response

Dear Editor,

We wish to thank both referees for their constructive criticisms. We have revised our manuscript to accommodate as much as possible their suggestions. We feel that, thanks to the referees’ suggestions, the manuscript is now highly improved and ready for the audience of Biomolecules. Our responses are as follows:

  • Referee #2

In line 44 & 45 they state that ATP/ADP is bound to 1 or 2 Mg cations while in Fig 1 they show 2 Mg cations. Is there still uncertainty about this?

Indeed, there are two Mg2+ binding sites, one required for catalysis and one associated with inhibition. In order to stabilize serine/threonine-specific protein kinases, metal ions are required in concentrations high enough to occupy both metal-binding sites. in other words, the number of magnesium ions depends on concentrations of MG2+. (the reference is added)

 In line 100 they should add " in an attempt' between Thr-288 and to. →Done

 In line 93 they should change This to These. They should give a reference when they first mention MLN 8084 & CD 532. →Done

Round 2

Reviewer 1 Report

The authors have addressed of most of my concerns. The following is one more suggestion authors need to consider.

In the cluster analysis (Figure S2), how many clusters were obtained for each structure?  Authors should analyze more sampled conformations (for example, at least top 3 clusters or even more) rather than a conformation from only the first cluster for each structure. As shown in Figure S3, the conformation of A-loop can dramatically change during the MD simulation. It would be interesting to see if its conformation could switch between open and close states in any structure during the simulation.

Author Response

Rome, 7/04/2021

Dear Editor,

We wish to thank both referees for their constructive criticisms. We have revised our manuscript to accommodate as much as possible of their suggestions. We feel that thanks to the referee’s suggestions, the manuscript is now highly improved and ready for the audience of Biomolecules. Our responses are as follows:

  • Referee #1

The authors have addressed most of my concerns. The following is one more suggestion authors need to consider.

In the cluster analysis (Figure S2), how many clusters were obtained for each structure?  Authors should analyze more sampled conformations (for example, at least the top 3 clusters or even more) rather than a conformation from only the first cluster for each structure. As shown in Figure S3, the conformation of the A-loop can dramatically change during the MD simulation. It would be interesting to see if its conformation could switch between open and close states in any structure during the simulation.

We have inserted an additional Table S2c that reports the statistics of the clusters analysis for the first three main clusters of all Aurora-A systems together with the number of cluster members and the number of cluster transitions for each cluster. Accordingly, we have modified the Methods and Results section to show the statistics of the cluster analysis. Actually, we cannot see open and closed states during the simulations of the different Aurora systems. What we have seen is that despite the flexibility of the A-loop as measured in the cluster analysis and the rate of transitions among the clusters, the centroids shown in figure S2a and S2b show that the loops are stable in their state despite some local flexibility and conformational variability (as evidence for examples in the case of MLN8054). 
